# Adaptive whitening with fast gain modulation and slow synaptic plasticity

**Lyndon R. Duong**[1,2]    **Eero P. Simoncelli**[1,2]    **Dmitri B. Chklovskii**[1,3]    **David Lipshutz**[1]

[1] Center for Computational Neuroscience, Flatiron Institute
[2] Center for Neural Science, New York University
[3] Neuroscience Institute, NYU Langone Medical School

{lyndon.duong, eero.simoncelli}@nyu.edu
{dchklovskii, dlipshutz}@flatironinstitute.org

## Abstract

Neurons in early sensory areas rapidly adapt to changing sensory statistics, both by normalizing the variance of their individual responses and by reducing correlations between their responses. Together, these transformations may be viewed as an adaptive form of statistical whitening. Existing mechanistic models of adaptive whitening exclusively use either synaptic plasticity or gain modulation as the biological substrate for adaptation; however, on their own, each of these models has significant limitations. In this work, we unify these approaches in a normative multi-timescale mechanistic model that adaptively whitens its responses with complementary computational roles for synaptic plasticity and gain modulation. Gains are modified on a fast timescale to adapt to the current statistical context, whereas synapses are modified on a slow timescale to match structural properties of the input statistics that are invariant across contexts. Our model is derived from a novel multi-timescale whitening objective that factorizes the inverse whitening matrix into basis vectors, which correspond to synaptic weights, and a diagonal matrix, which corresponds to neuronal gains. We test our model on synthetic and natural datasets and find that the synapses learn optimal configurations over long timescales that enable adaptive whitening on short timescales using gain modulation.

## 1 Introduction

Individual neurons in early sensory areas rapidly adapt to changing sensory statistics by normalizing the variance of their responses [1–3]. At the population level, neurons also adapt by reducing correlations between their responses [4; 5]. These adjustments enable the neurons to maximize the information that they transmit by utilizing their entire dynamic range and reducing redundancies in their representations [6–9]. A natural normative interpretation of these transformations is *adaptive whitening*, a context-dependent linear transformation of the sensory inputs yielding responses that have unit variance and are uncorrelated.

Decorrelation of the neural responses requires coordination between neurons and the neural mechanisms underlying such coordination are not known. Since neurons communicate via synaptic connections, it is perhaps unsurprising that most existing mechanistic models of adaptive whitening decorrelate neural responses by modifying the strength of these connections [10–16]. However, long-term synaptic plasticity is generally associated with long-term learning and memory [17], and thus may not be a suitable biological substrate for adaptive whitening (though short-term synaptic

37th Conference on Neural Information Processing Systems (NeurIPS 2023).

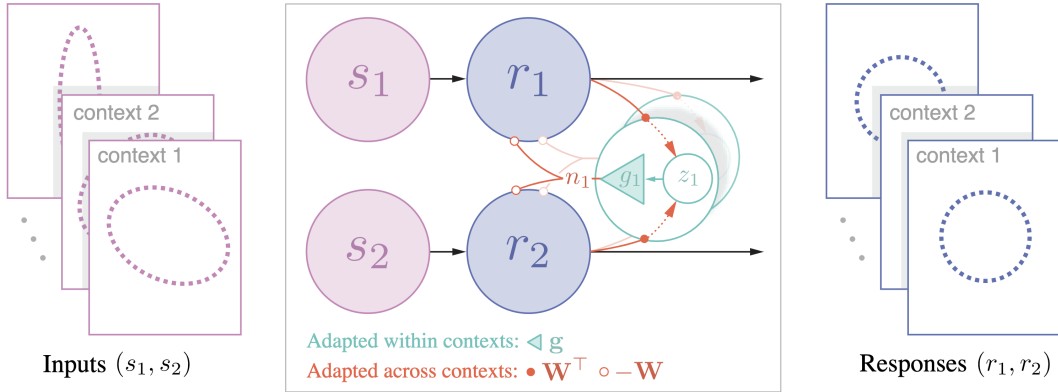

Figure 1: Adaptive whitening circuit, illustrated with $N = 2$ primary neurons and $K = 2$ interneurons. **Left**: Dashed ellipses representing the covariance matrices of 2D stimuli $\mathbf{s}$ drawn from different statistical contexts. **Center**: Primary neurons (shaded blue circles) receive feedforward stimulus inputs (shaded purple circles), $\mathbf{s}$, and recurrent weighted inputs, $-\mathbf{Wn}$, from the interneurons (teal circles), producing responses $\mathbf{r}$. The interneurons receive weighted inputs, $\mathbf{z} = \mathbf{W}^\top \mathbf{r}$, from the primary neurons, which are then multiplied elementwise by gains $\mathbf{g}$ to generate their outputs, $\mathbf{n} = \mathbf{g} \circ \mathbf{z}$. The gains $\mathbf{g}$ are modulated at a fast timescale to adaptively whiten within a specific stimulus context. Concurrently, the synaptic weights are optimized at a slower timescale to learn structural properties of the inputs across contexts. **Right:** Dashed unit circles representing the whitened circuit responses $\mathbf{r}$ in each statistical context.

plasticity has been reported [18]). On the other hand, there is extensive neuroscience literature on rapid and reversible gain modulation [19–26]. Motivated by this, Duong et al. [27] proposed a mechanistic model of adaptive whitening in a neural circuit with *fixed* synaptic connections that adapts exclusively by modifying the gains of interneurons that mediate communication between the primary neurons. They demonstrate that an appropriate choice of the fixed synaptic weights can both accelerate adaptation and significantly reduce the number of interneurons that the circuit requires. However, it remains unclear how the circuit *learns* such an optimal synaptic configuration, which would seem to require synaptic plasticity.

In this study, we combine the learning and adaptation of synapses and gains, respectively, in a unified mechanistic neural circuit model that adaptively whitens its inputs over multiple timescales (Fig. 1). Our main contributions are as follows:

1. We introduce a novel adaptive whitening objective in which the (inverse) whitening matrix is factorized into a synaptic weight matrix that is optimized across contexts and a diagonal (gain) matrix that is optimized within each statistical context.

2. With this objective, we derive a multi-timescale online algorithm for adaptive whitening that can be implemented in a neural circuit comprised of primary neurons and an auxiliary population of interneurons with slow synaptic plasticity and fast gain modulation (Fig. 1).

3. We test our algorithm on synthetic and natural datasets, and demonstrate that the synapses learn optimal configurations over long timescales that enable the circuit to adaptively whiten its responses on short timescales exclusively using gain modulation.

Beyond the biological setting, multi-timescale learning and adaptation may also prove important in machine learning systems. For example, Mohan et al. [28] introduced "gain-tuning", in which the gains of channels in a deep denoising neural network (with pre-trained synaptic weights) are adjusted to improve performance on samples with out-of-distribution noise corruption or signal properties. The normative multi-timescale framework developed here offers a new approach to continual learning and test-time adaptation problems such as this.

## 2 Adaptive symmetric whitening

Consider a neural population with $N$ primary neurons (Fig. 1). The stimulus inputs to the primary neurons are represented by a random $N$-dimensional vector $\mathbf{s}$ whose distribution $p(\mathbf{s}|c)$ depends

on a latent context variable $c$. The stimulus inputs $\mathbf{s}$ can be inputs to peripheral sensory neurons (e.g., the rates at which photons are absorbed by $N$ cones) or inputs to neurons in an early sensory area (e.g., glomerulus inputs to $N$ mitral cells in the olfactory bulb). Context variables can include location (e.g., a forest or a meadow) or time (e.g., season or time of day). For simplicity, we assume the context-dependent inputs are centered; that is, $\mathbb{E}_{\mathbf{s} \sim p(\mathbf{s}|c)}[\mathbf{s}] = \mathbf{0}$, where $\mathbb{E}_{\mathbf{s} \sim p(\mathbf{s}|c)}[\cdot]$ denotes the expectation over the conditional distribution $p(\mathbf{s}|c)$ and $\mathbf{0}$ denotes the vector of zeros. See Appx. A for a consolidated list of notation used throughout this work.

The goal of adaptive whitening is to linearly transform the inputs $\mathbf{s}$ so that, conditioned on the context variable $c$, the $N$-dimensional neural responses $\mathbf{r}$ have identity covariance matrix; that is,

$$\mathbf{r} = \mathbf{F}_c \mathbf{s} \qquad \text{such that} \qquad \mathbb{E}_{\mathbf{s} \sim p(\mathbf{s}|c)}\left[\mathbf{r}\mathbf{r}^\top\right] = \mathbf{I}_N,$$

where $\mathbf{F}_c$ is a context-dependent $N \times N$ whitening matrix. Whitening is not a unique transformation—left multiplication of the whitening matrix $\mathbf{F}_c$ by any $N \times N$ orthogonal matrix results in another whitening matrix. We focus on symmetric whitening, also referred to as Zero-phase Components Analysis (ZCA) whitening or Mahalanobis whitening, in which the whitening matrix for context $c$ is uniquely defined as

$$\mathbf{F}_c = \mathbf{C}_{ss}^{-1/2}(c), \qquad\qquad \mathbf{C}_{ss}(c) := \mathbb{E}_{\mathbf{s} \sim p(\mathbf{s}|c)}\left[\mathbf{s}\mathbf{s}^\top\right], \qquad\qquad (1)$$

where we assume $\mathbf{C}_{ss}(c)$ is positive definite for all contexts $c$. This is the unique whitening transformation that minimizes the mean-squared difference between the inputs and the outputs [29].

To derive an algorithm that learns the symmetric whitening matrix $\mathbf{F}_c$, we express $\mathbf{F}_c$ as the solution to an appropriate optimization problem, which is similar to the optimization problem in [15, top of page 6]. For a context $c$, we can write the *inverse* symmetric whitening matrix $\mathbf{M}_c := \mathbf{F}_c^{-1}$ as the unique optimal solution to the minimization problem

$$\mathbf{M}_c = \underset{\mathbf{M} \in \mathbb{S}_{++}^N}{\arg\min}\, f_c(\mathbf{M}), \qquad\qquad f_c(\mathbf{M}) := \mathrm{Tr}\left(\mathbf{M}^{-1}\mathbf{C}_{ss}(c) + \mathbf{M}\right), \qquad\qquad (2)$$

where $\mathbb{S}_{++}^N$ denotes the set of $N \times N$ positive definite matrices.[1] This follows from the fact that $f_c(\mathbf{M})$ is strictly convex with its unique minimum achieved at $\mathbf{M}_c$, where $f_c(\mathbf{M}_c) = 2\,\mathrm{Tr}(\mathbf{M}_c)$ (Appx. B.1). Existing recurrent neural circuit models of adaptive whitening solve the minimization problem in Eq. 2 by choosing a matrix factorization of $\mathbf{M}_c$ and then optimizing the components [12; 13; 15; 27].

## 3 Adaptive whitening in neural circuits: a matrix factorization perspective

Here, we review two adaptive whitening objectives, which we then unify into a single objective that adaptively whitens responses across multiple timescales.

### 3.1 Objective for adaptive whitening via synaptic plasticity

Pehlevan and Chklovskii [12] proposed a recurrent neural circuit model that whitens neural responses by adjusting the synaptic weights between the $N$ primary neurons and $K \geq N$ auxiliary interneurons according to a Hebbian update rule. Their circuit can be derived by factorizing the context-dependent matrix $\mathbf{M}_c$ into a symmetric product $\mathbf{M}_c = \mathbf{W}_c \mathbf{W}_c^\top$ for some context-dependent $N \times K$ matrix $\mathbf{W}_c$ [15]. Substituting this factorization into Eq. 2 results in the synaptic plasticity objective in Table 1. In the recurrent circuit implementation, $\mathbf{W}_c^\top$ denotes the feedforward weight matrix of synapses connecting primary neurons to interneurons and the matrix $-\mathbf{W}_c$ denotes the feedback weight matrix of synapses connecting interneurons to primary neurons. Importantly, under this formulation, the circuit may reconfigure both the synaptic connections and synaptic strengths each time the context $c$ changes, which runs counter to the prevailing view that synaptic plasticity implements long-term learning and memory [17].

### 3.2 Objective for adaptive whitening via gain modulation

Duong et al. [27] proposed a neural circuit model with *fixed* synapses that whitens the $N$ primary responses by adjusting the multiplicative gains in a set of $K$ auxiliary interneurons. To derive a neural

---

[1]For technical purposes, we extend the definition of $f_c$ to all $\mathbb{S}^N$ by setting $f_c(\mathbf{M}) = \infty$ if $\mathbf{M} \notin \mathbb{S}_{++}^N$.

Table 1: Factorizations of the inverse whitening matrix $\mathbf{M}_c$ and objectives for adaptive whitening circuits.

| Model | Matrix factorization | Objective |
|---|---|---|
| Synaptic plasticity [12] | $\mathbf{W}_c\mathbf{W}_c^\top$ | $\min_{\mathbf{W}} f_c\left(\mathbf{W}\mathbf{W}^\top\right)$ |
| Gain modulation [27] | $\mathbf{I}_N + \mathbf{W}_{\text{fix}}\text{diag}(\mathbf{g}_c)\mathbf{W}_{\text{fix}}^\top$ | $\min_{\mathbf{g}} f_c\left(\mathbf{I}_N + \mathbf{W}_{\text{fix}}\text{diag}(\mathbf{g})\mathbf{W}_{\text{fix}}^\top\right)$ |
| Multi-timescale (ours) | $\alpha\mathbf{I}_N + \mathbf{W}\text{diag}(\mathbf{g}_c)\mathbf{W}^\top$ | $\min_{\mathbf{W}} \mathbb{E}_{c\sim p(c)}\left[\min_{\mathbf{g}} f_c\left(\alpha\mathbf{I}_N + \mathbf{W}\text{diag}(\mathbf{g})\mathbf{W}^\top\right)\right]$ |

circuit with gain modulation, they considered a novel diagonalization of the inverse whitening matrix, $\mathbf{M}_c = \mathbf{I}_N + \mathbf{W}_{\text{fix}}\text{diag}(\mathbf{g}_c)\mathbf{W}_{\text{fix}}^\top$, where $\mathbf{W}_{\text{fix}}$ is an arbitrary, but fixed $N \times K$ matrix of synaptic weights (with $K \geq K_N := N(N+1)/2$) and $\mathbf{g}_c$ is an adaptive, context-dependent real-valued $K$-dimensional vector of gains. Note that unlike the conventional eigen-decomposition, the number of elements along the diagonal matrix is significantly larger than the dimensionality of the input space. Substituting this factorization into Eq. 2 results in the gain modulation objective in Table 1. As in the synaptic plasticity model, $\mathbf{W}_{\text{fix}}^\top$ denotes the weight matrix of synapses connecting primary neurons to interneurons while $-\mathbf{W}_{\text{fix}}$ connects interneurons to primary neurons. In contrast to the synaptic plasticity model, the interneuron outputs are modulated by context-dependent multiplicative gains, $\mathbf{g}_c$, that are adaptively adjusted to whiten the circuit responses.

Duong et al. [27] demonstrate that an appropriate choice of the fixed synaptic weight matrix can both accelerate adaptation and significantly reduce the number of interneurons in the circuit. In particular, the gain modulation circuit can whiten *any* input distribution provided the gains vector $\mathbf{g}_c$ has dimension $K \geq K_N$ (the number of degrees of freedom in an $N \times N$ symmetric covariance matrix). However, in practice, the circuit need only adapt to input distributions corresponding to *natural* input statistics [7; 30–32]. For example, the statistics of natural images are approximately translation-invariant, which significantly reduces the degrees of freedom in their covariance matrices, from $\mathcal{O}(N^2)$ to $\mathcal{O}(N)$. Therefore, while the space of all possible correlation structures is $K_N$-dimensional, the set of natural statistics likely has far fewer degrees of freedom and an optimal selection of the weight matrix $\mathbf{W}_{\text{fix}}$ can potentially offer dramatic reductions in the number of interneurons $K$ required to adapt. As an example, Duong et al. [27] specify a weight matrix for performing "local" whitening with $\mathcal{O}(N)$ interneurons when the input correlations are spatially-localized (e.g., as in natural images). However, they do not prescribe a method for *learning* a (synaptic) weight matrix that is optimal across the set of natural input statistics.

### 3.3 Unified objective for adaptive whitening via synaptic plasticity and gain modulation

We unify and generalize the two disparate adaptive whitening approaches [12; 27] in a single *multi-timescale* nested objective in which gains $\mathbf{g}$ are optimized within each context and synaptic weights $\mathbf{W}$ are optimized across contexts. In particular, we optimize, with respect to $\mathbf{W}$, the expectation of the objective from [27] (for some fixed $K \geq 1$) over the distribution of contexts $p(c)$:

$$\min_{\mathbf{W}\in\mathbb{R}^{N\times K}} \mathbb{E}_{c\sim p(c)}\left[\min_{\mathbf{g}\in\mathbb{R}^K} f_c\left(\alpha\mathbf{I}_N + \mathbf{W}\text{diag}(\mathbf{g})\mathbf{W}^\top\right)\right], \qquad (3)$$

where we have also generalized the objective from [27] by including a fixed multiplicative factor $\alpha \geq 0$ in front of the identity matrix $\mathbf{I}_N$, and we have relaxed the requirement that $K \geq K_N$.

What is an optimal solution of Eq. 3? Since the convex function $f_c$ is uniquely minimized at $\mathbf{M}_c$, a sufficient condition for the optimality of a synaptic weight matrix $\mathbf{W}$ is that for each context $c$, there is a gain vector $\mathbf{g}_c$ such that $\alpha\mathbf{I}_N + \mathbf{W}\text{diag}(\mathbf{g}_c)\mathbf{W}^\top = \mathbf{M}_c$. Importantly, under such a synaptic configuration, the function $f_c$ can attain its minimum exclusively by adjusting the gains vector $\mathbf{g}$. In the space of covariance matrices, we can express the statement as

$$\mathbf{C}_{ss}(c) \in \mathbb{F}(\mathbf{W}) := \left\{\left[\alpha\mathbf{I}_N + \mathbf{W}\text{diag}(\mathbf{g})\mathbf{W}^\top\right]^2 : \mathbf{g} \in \mathbb{R}^K\right\} \cap \mathbb{S}_{++}^N \quad \text{for every context } c,$$

where $\mathbb{F}(\mathbf{W})$ contains the set of covariance matrices that can be whitened with fixed synapses $\mathbf{W}$ and adaptive gains $\mathbf{g}$. Fig. 2 provides an intuitive Venn diagram comparing a non-optimal synaptic configuration $\mathbf{W}_0$ and an optimal synaptic configuration $\mathbf{W}_T$.

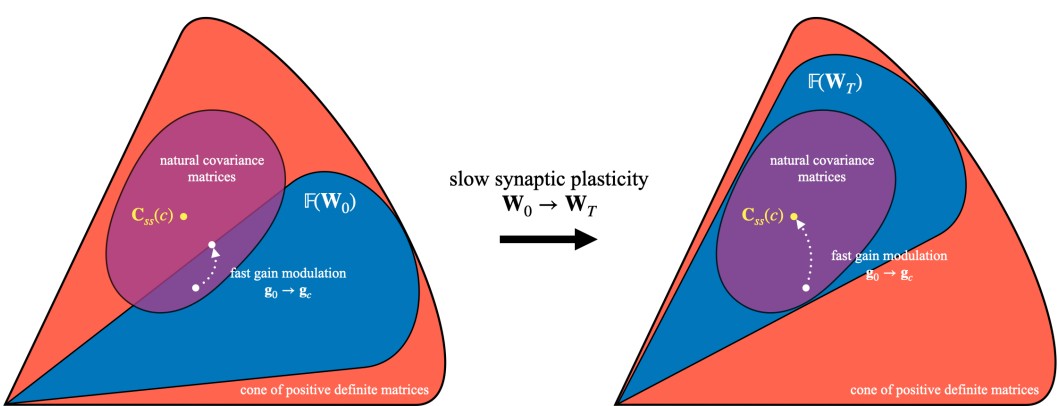

Figure 2: Illustration of multi-timescale learning in the space of covariance matrices. Orange and purple regions (identical on the left and right) respectively represent the cone of all positive definite matrices $\mathbb{S}_{++}^N$, and the subset of naturally-occurring covariance matrices $\{\mathbf{C}_{ss}(c)\}$. Blue regions represent the set of covariance matrices that can be whitened with adaptive gains for a particular synaptic weight matrix. On each side, the yellow circle denotes a naturally-occurring input covariance matrix $\mathbf{C}_{ss}(c)$ and the dotted white curve illustrates the trajectory of covariance matrices the circuit is adapted to whiten as the gains are modulated (with fixed synapses, note the dotted white curve remains in the blue region). **Left:** With initial synaptic weights $\mathbf{W}_0$ the circuit cannot whiten some natural input distributions exclusively via gain modulation, i.e., $\{\mathbf{C}_{ss}(c)\} \not\subset \mathbb{F}(\mathbf{W}_0)$. **Right:** After learning optimal synaptic weights $\mathbf{W}_T$, the circuit can match any naturally-occurring covariance matrix using gain modulation, i.e., $\{\mathbf{C}_{ss}(c)\} \subset \mathbb{F}(\mathbf{W}_T)$.

## 4  Multi-timescale adaptive whitening algorithm and circuit implementation

In this section, we derive an online algorithm for optimizing the multi-timescale objective in Eq. 3, then map the algorithm onto a neural circuit with fast gain modulation and slow synaptic plasticity. Direct optimization of Eq. 3 results in an offline (or full batch) algorithm that requires the network to have access to the covariance matrices $\mathbf{C}_{ss}(c)$, Appx. C. Therefore, to derive an online algorithm that includes neural dynamics, we first add neural responses $\mathbf{r}$ to the objective, which introduces a third timescale to the objective. We then derive a multi-timescale gradient-based algorithm for optimizing the objective.

**Adding neural responses to the objective.**     First, observe that we can write $f_c(\mathbf{M})$, for $\mathbf{M} \in \mathbb{S}_{++}^N$, in terms of the neural responses $\mathbf{r}$:

$$f_c(\mathbf{M}) = \mathbb{E}_{\mathbf{s} \sim p(\mathbf{s}|c)} \left[ \max_{\mathbf{r} \in \mathbb{R}^N} \mathrm{Tr} \left( 2\mathbf{r}\mathbf{s}^\top - \mathbf{M}\mathbf{r}\mathbf{r}^\top + \mathbf{M} \right) \right]. \tag{4}$$

To see this, maximize over $\mathbf{r}$ to obtain $\mathbf{r} = \mathbf{M}^{-1}\mathbf{s}$ and then use the definition of $\mathbf{C}_{ss}(c)$ from Eq. 1 (Appx. B.2). Substituting this expression for $f_c$, with $\mathbf{M} = \alpha \mathbf{I}_N + \mathbf{W}\mathrm{diag}(\mathbf{g})\mathbf{W}^\top$, into Eq. 3, dropping the constant term $\alpha \mathbf{I}_N$ term and using the cyclic property of the trace operator results in the following objective with 3 nested optimizations (Appx. B.2):

$$\min_{\mathbf{W} \in \mathbb{R}^{N \times K}} \mathbb{E}_{c \sim p(c)} \left[ \min_{\mathbf{g} \in \mathbb{R}^K} \mathbb{E}_{\mathbf{s} \sim p(\mathbf{s}|c)} \left[ \max_{\mathbf{r} \in \mathbb{R}^N} \ell(\mathbf{W}, \mathbf{g}, \mathbf{r}, \mathbf{s}) \right] \right], \tag{5}$$

$$\text{where} \quad \ell(\mathbf{W}, \mathbf{g}, \mathbf{r}, \mathbf{s}) := 2\mathbf{r}^\top \mathbf{s} - \alpha \|\mathbf{r}\|^2 - \sum_{i=1}^K g_i \left[ (\mathbf{w}_i^\top \mathbf{r})^2 - \|\mathbf{w}_i\|^2 \right].$$

The inner-most optimization over $\mathbf{r}$ corresponds to neural responses and will lead to recurrent neural dynamics. The outer 2 optimizations correspond to the optimizations over the gains $\mathbf{g}$ and synaptic weights $\mathbf{W}$ from Eq. 3.

To solve Eq. 5 in the online setting, we assume there is a timescale separation between neural dynamics and the gain/weight updates. This allows us to perform the optimization over $\mathbf{r}$ before optimizing $\mathbf{g}$ and $\mathbf{W}$ concurrently. This is biologically sensible: neural responses (e.g., action potential firing) operate on a much faster timescale than gain modulation and synaptic plasticity [25; 33].

**Recurrent neural dynamics.** At each iteration, the circuit receives a stimulus $\mathbf{s}$. We maximize $\ell(\mathbf{W}, \mathbf{g}, \mathbf{r}, \mathbf{s})$ with respect to $\mathbf{r}$ by iterating the following gradient-ascent steps that correspond to repeated timesteps of the recurrent circuit (Fig. 1) until the responses equilibrate:

$$\mathbf{r} \leftarrow \mathbf{r} + \eta_r \left( \mathbf{s} - \sum_{i=1}^{K} n_i \mathbf{w}_i - \alpha \mathbf{r} \right), \tag{6}$$

where $\eta_r > 0$ is a small constant, $z_i = \mathbf{w}_i^\top \mathbf{r}$ denotes the weighted input to the $i^{\text{th}}$ interneuron, $n_i = g_i z_i$ denotes the gain-modulated output of the $i^{\text{th}}$ interneuron. For each $i$, synaptic weights, $\mathbf{w}_i$, connect the primary neurons to the $i^{\text{th}}$ interneuron and symmetric weights, $-\mathbf{w}_i$, connect the $i^{\text{th}}$ interneuron to the primary neurons. From Eq. 6, we see that the neural responses are driven by feedforward stimulus inputs $\mathbf{s}$, recurrent weighted feedback from the interneurons $-\mathbf{W}\mathbf{n}$, and a leak term $-\alpha \mathbf{r}$.

**Fast gain modulation and slow synaptic plasticity.** After the neural activities equilibrate, we minimize $\ell(\mathbf{W}, \mathbf{g}, \mathbf{r}, \mathbf{s})$ by taking concurrent gradient-descent steps

$$\Delta g_i = \eta_g \left( z_i^2 - \|\mathbf{w}_i\|^2 \right) \tag{7}$$

$$\Delta \mathbf{w}_i = \eta_w \left( \mathbf{r} n_i - \mathbf{w}_i g_i \right), \tag{8}$$

where $\eta_g$ and $\eta_w$ are the respective learning rates for the gains and synaptic weights. By choosing $\eta_g \gg \eta_w$, we can ensure that the gains are updated at a faster timescale than the synaptic weights.

The update to the $i^{\text{th}}$ interneuron's gain $g_i$ depends on the difference between the online estimate of the variance of its input, $z_i^2$, and the squared-norm of the $i^{\text{th}}$ synaptic weight vector, $\|\mathbf{w}_i\|^2$, quantities that are both locally available to the $i^{\text{th}}$ interneuron. Using the fact that $z_i = \mathbf{w}_i^\top \mathbf{r}$, we can rewrite the gain update as $\Delta g_i = \eta_g [\mathbf{w}_i^\top (\mathbf{r}\mathbf{r}^\top - \mathbf{I}_N) \mathbf{w}_i]$. From this expression, we see that the gains equilibrate when the marginal variance of the responses along the direction $\mathbf{w}_i$ is 1, for $i = 1, \dots, K$.

The update to the $(i,j)^{\text{th}}$ synaptic weight $w_{ij}$ is proportional to the difference between $r_i n_j$ and $w_{ij} g_j$, which depends only on variables that are available in the pre- and postsynaptic neurons. Since $r_i n_j$ is the product of the pre- and postsynaptic activities, we refer to this update as *Hebbian*. In Appx. E.2, we decouple the feedforward weights $\mathbf{w}_i^\top$ and feedback weights $-\mathbf{w}_i$ and provide conditions under which the symmetry asymptotically holds.

**Multi-timescale online algorithm.** Combining the neural dynamics, gain modulation and synaptic plasticity yields our online multi-timescale adaptive whitening algorithm, which we express in vector-matrix form with '$\circ$' denoting the Hadamard (elementwise) product of two vectors:

---

**Algorithm 1:** Multi-timescale adaptive whitening via synaptic plasticity and gain modulation

---
1: **Input:** $\mathbf{s}_1, \mathbf{s}_2, \dots \in \mathbb{R}^N$
2: **Initialize:** $\mathbf{W} \in \mathbb{R}^{N \times K}$; $\mathbf{g} \in \mathbb{R}^K$; $\eta_r > 0$; $\eta_g \gg \eta_w > 0$
3: **for** $t = 1, 2, \dots$ **do**
4:     $\mathbf{r}_t \leftarrow \mathbf{0}$
5:     **while** not converged **do**
6:         $\mathbf{z}_t \leftarrow \mathbf{W}^\top \mathbf{r}_t$ ;                `// interneuron inputs`
7:         $\mathbf{n}_t \leftarrow \mathbf{g} \circ \mathbf{z}_t$ ;        `// gain-modulated interneuron outputs`
8:         $\mathbf{r}_t \leftarrow \mathbf{r}_t + \eta_r (\mathbf{s}_t - \mathbf{W}\mathbf{n}_t - \alpha \mathbf{r}_t)$ ;        `// recurrent neural dynamics`
9:     **end while**
10:    $\mathbf{g} \leftarrow \mathbf{g} + \eta_g \left( \mathbf{z}_t \circ \mathbf{z}_t - \text{diag} \left( \mathbf{W}^\top \mathbf{W} \right) \right)$ ;        `// gains update`
11:    $\mathbf{W} \leftarrow \mathbf{W} + \eta_w \left( \mathbf{r}_t \mathbf{n}_t^\top - \mathbf{W}\text{diag}(\mathbf{g}) \right)$ ;     `// synaptic weights update`
12: **end for**

---

Alg. 1 is naturally viewed as a *unification* and generalization of previously proposed neural circuit models for adaptation. When $\alpha = 0$ and the gains $\mathbf{g}$ are constant (e.g., $\eta_g = 0$) and identically equal to the vector of ones $\mathbf{1}$ (so that $\mathbf{n}_t = \mathbf{z}_t$), we recover the synaptic plasticity algorithm from [15]. Similarly, when $\alpha = 1$ and the synaptic weights $\mathbf{W}$ are fixed (e.g., $\eta_w = 0$), we recover the gain modulation algorithm from [27].

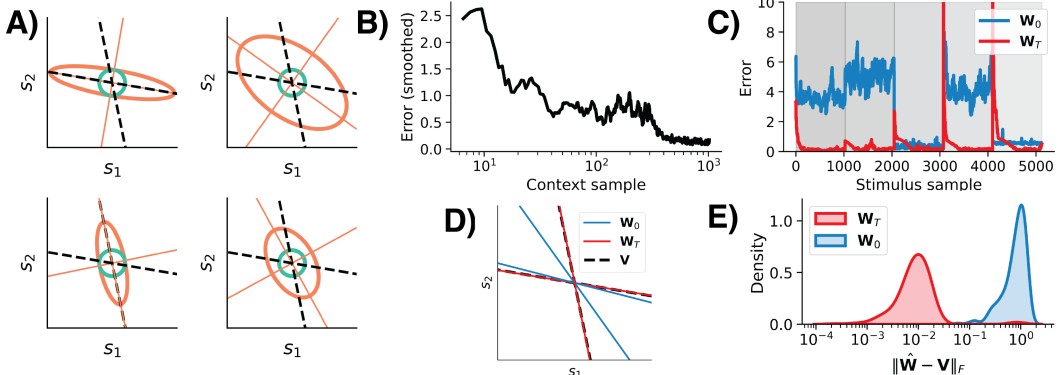

Figure 3: Adaptive whitening of a synthetic dataset with $N = 2$, $\eta_w = $ 1E-5, $\eta_g = $ 5E-2. **A)** Covariance ellipses (orange) of 4 out of 64 synthesized contexts. Black dashed lines are axes corresponding to the column vectors of $\mathbf{V}$. The unit circle is shown in green. Since the column vectors of $\mathbf{V}$ are not orthogonal, these covariance matrices do *not* share a common set of eigenvectors (orange lines). **B)** Whitening error with a moving average window spanning the last 10 contexts. **C)** Error at each stimulus presentation within five different contexts (gray panels), presented with $\mathbf{W}_0$, or $\mathbf{W}_T$. **D)** Column vectors of $\mathbf{W}_0$, $\mathbf{W}_T$, $\mathbf{V}$ (each axis corresponds to the span of one column vector in $\mathbb{R}^2$). **E)** Smoothed distributions of error (in Frobenius norm) between $\hat{\mathbf{W}}$ and $\mathbf{V}$ across 250 random initializations of $\mathbf{W}_0$.

## 5 Numerical experiments

We test Alg. 1 on stimuli $\mathbf{s}_1, \mathbf{s}_2, \ldots$ drawn from slowly fluctuating latent contexts $c_1, c_2, \ldots$; that is, $\mathbf{s}_t \sim p(\mathbf{s}|c_t)$ and $c_t = c_{t-1}$ with high probability.[2] To measure performance, we evaluate the operator norm on the difference between the expected response covariance and the identity matrix:

$$\text{Error}(t) = \|\mathbf{M}_t^{-1}\mathbf{C}_{ss}(c_t)\mathbf{M}_t^{-1} - \mathbf{I}_N\|_{\text{op}}, \qquad \mathbf{M}_t := \alpha\mathbf{I}_N + \mathbf{W}_t\text{diag}(\mathbf{g})\mathbf{W}_t^\top. \qquad (9)$$

Geometrically, this "worst-case" error measures the maximal Euclidean distance between the ellipsoid corresponding to $\mathbf{M}_t^{-1}\mathbf{C}_{ss}(c_t)\mathbf{M}_t^{-1}$ and the $(N-1)$-sphere along all possible axes. To compare two synaptic weight matrices $\mathbf{A}, \mathbf{B} \in \mathbb{R}^{N \times K}$, we evaluate $\|\hat{\mathbf{A}} - \mathbf{B}\|_F$, where $\hat{\mathbf{A}} = \mathbf{AP}$ and $\mathbf{P}$ is the permutation matrix (with possible sign flips) that minimizes the error.

### 5.1 Synthetic dataset

To validate our model, we first consider a 2-dimensional synthetic dataset in which an optimal solution is known. Suppose that each context-dependent inverse whitening matrix is of the form $\mathbf{M}_c = \mathbf{I}_N + \mathbf{V}\mathbf{\Lambda}(c)\mathbf{V}^\top$, where $\mathbf{V}$ is a fixed $2 \times 2$ matrix and $\mathbf{\Lambda}(c) = \text{diag}(\lambda_1(c), \lambda_2(c))$ is a context-dependent diagonal matrix. Then, in the case $\alpha = 1$ and $K = 2$, an optimal solution of the objective in Eq. 3 is when the column vectors of $\mathbf{W}$ align with the column vectors of $\mathbf{V}$.

To generate this dataset, we chose the column vectors of $\mathbf{V}$ uniformly from the unit circle, so they are *not* generally orthogonal. For each context $c = 1, \ldots, 64$, we assume the diagonal entries of $\mathbf{\Lambda}(c)$ are sparse and i.i.d.: with probability 1/2, $\lambda_i(c)$ is set to zero and with probability 1/2, $\lambda_i(c)$ is chosen uniformly from the interval $[0, 4]$. Example covariance matrices from different contexts are shown in Fig. 3A (note that they do *not* share a common eigen-decomposition). Finally, for each context, we generate 1E3 i.i.d. samples with context-dependent distribution $\mathbf{s} \sim \mathcal{N}(\mathbf{0}, \mathbf{M}_c^2)$.

We test Alg. 1 with $\alpha = 1$, $K = 2$, $\eta_w = $ 1E-5, and $\eta_g = $ 5E-2 on these sequences of synthetic inputs with the column vectors of $\mathbf{W}_0$ chosen uniformly from the unit circle. The model successfully learns to whiten the different contexts, as indicated by the decreasing whitening error with the number of contexts presented (Fig. 3B). At the end of training, the synaptic weight matrix $\mathbf{W}_T$ is optimized such that the circuit can adapt to changing contexts exclusively by adjusting its gains. This is evidenced by the fact that when the context changes, there is a brief spike in error as the gains adapt to the new context (Fig. 3C, red line). By contrast, the error remains high in many of the contexts when using

---

[2]Python code accompanying this study can be found at https://github.com/lyndond/multi_timescale_whitening.

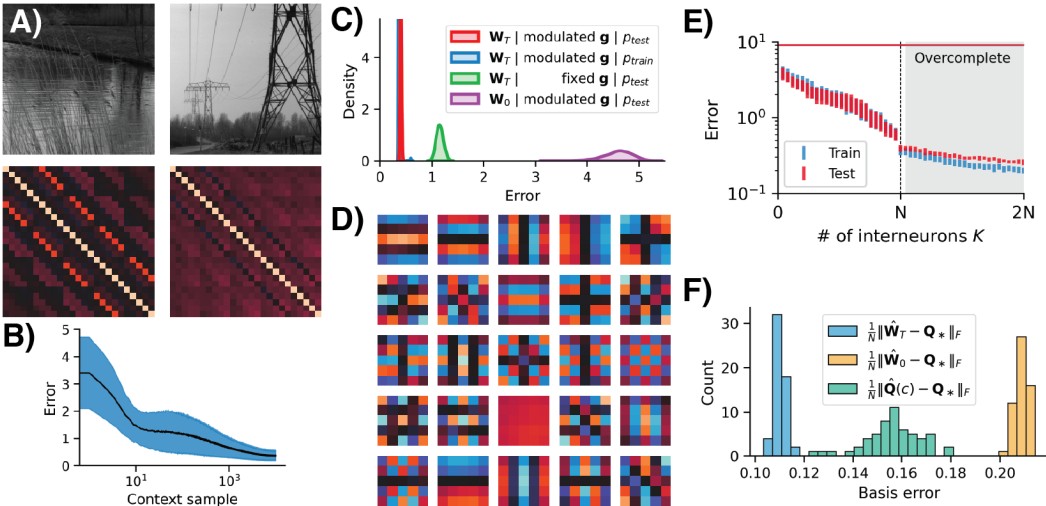

Figure 4: Adaptive whitening of natural images. **A)** Examples of 2 out of 56 high-resolution images (top) with each image corresponding to a separate context. For each image, $5 \times 5$ pixel patches are randomly sampled to generate context-dependent stimuli with covariance matrix $\mathbf{C}_{ss}(c) \in \mathbb{S}^{25}_{++}$ (bottom). **B)** Mean error during training (Eq. 9) with $K = N = 25$. Shaded region is standard deviation over 2E3 random initializations $\mathbf{W}_0 \in O(25)$. **C)** Smoothed distributions of average adaptive whitening error over all 2E3 initializations. The red distribution corresponds to the error on the held-out images with fixed learned synapses $\mathbf{W}_T$ and modulated gains $\mathbf{g}$. The blue (resp. green, purple) distribution corresponds to the same error, but tested on the training images (resp. with fixed gains equal to the average gains over the final 100 iterations, with fixed random synapses $\mathbf{W}_0$). **D)** The learned weights (re-shaped columns of $\mathbf{W}_T$) approximate orthogonal 2D sinusoids. **E)** Final error (after $T = 5E4$ iterations) as a function of number of interneurons $K$. Bars are standard deviations centered on the mean error at each $K$. The red horizontal line denotes average error when $K = 0$ (in which case $\mathbf{r} = \mathbf{s}$). **F)** Frobenius norm between the eigenbasis of $\mathbb{E}_{c \sim p(c)}[\mathbf{C}_{ss}(c)]$ (i.e. across all contexts), $\mathbf{Q}_*$, with $\mathbf{W}_T$, $\mathbf{W}_0$, and the eigenbasis of each individual context covariance, $\mathbf{Q}(c)$, when $K = N = 25$. See Appx. D for additional experiments.

the initial random synaptic weight matrix $\mathbf{W}_0$ (Fig. 3C, blue line). In particular, the synapses learn (across contexts) an optimal configuration in the sense that the column vectors of $\mathbf{W}$ learn to align with the column vectors of $\mathbf{V}$ over the course of training (Fig. 3DE).

## 5.2 Natural images dataset

By hand-crafting a particular set of synaptic weights, Duong et al. [27] showed that their adaptive whitening network can approximately whiten a dataset of natural image patches with $\mathcal{O}(N)$ gain-modulating interneurons instead of $\mathcal{O}(N^2)$. Here, we show that our model can exploit spatial structure across natural scenes to *learn* an optimal set of synaptic weights by testing our algorithm on 56 high-resolution natural images [34] (Fig. 4A, top). For each image, which corresponds to a separate context $c$, $5 \times 5$ pixel image patches are randomly sampled and vectorized to generate context-dependent samples $\mathbf{s} \in \mathbb{R}^{25}$ with covariance matrix $\mathbf{C}_{ss}(c) \in \mathbb{S}^{25}_{++}$ (Fig. 4A, bottom). We train our algorithm in the offline setting where we have direct access to the context-dependent covariance matrices (Appx. C, Alg. 2, $\alpha = 1$, $J = 50$, $\eta_g$ =5E-1, $\eta_w$ =5E-2) with $K = N = 25$ and random $\mathbf{W}_0 \in O(25)$ on a training set of 50 of the images, presented uniformly at random 1E3 total times. We find that the model successfully learns a basis that enables adaptive whitening *across* different visual contexts via gain modulation, as shown by the decreasing training error (Eq. 9) in Fig. 4B.

How does the network learn to leverage statistical structure that is consistent across contexts? We test the circuit with fixed synaptic weights $\mathbf{W}_T$ and modulated (adaptive) gains $\mathbf{g}$ on stimuli from the held-out images (Fig. 4C, red distribution shows the smoothed error over 2E3 random initializations $\mathbf{W}_0$). The circuit performs as well on the held-out images as on the training images (Fig. 4C, red versus blue distributions). In addition, the circuit with learned synaptic weights $\mathbf{W}_T$ and modulated gains $\mathbf{g}$ outperforms the circuit with learned synaptic weights $\mathbf{W}_T$ and fixed gains (Fig. 4C, green distribution), and significantly outperforms the circuit with random synaptic weights $\mathbf{W}_0$ and modulated gains (Fig. 4C, purple distribution). Together, these results suggest that the circuit

learns features $\mathbf{W}_T$ that enable the circuit to adaptively whiten across statistical contexts exclusively using gain modulation, and that gain modulation is crucial to the circuit's ability to adaptively whiten. In Fig. 4D, we visualize the learned filters (columns of $\mathbf{W}_T$), and find that they are approximately equal to the 2D discrete cosine transform (DCT, Appx. D), an orthogonal basis that is known to approximate the eigenvectors of natural image patch covariances [35; 36].

To examine the dependence of performance on the number of interneurons $K$, we train the algorithm with $K = 1, \ldots, 2N$ and report the final error in Fig. 4E. There is a steady drop in error as $K$ ranges from 1 to $N$, at which point there is a (discontinuous) drop in error followed by a continued, but more gradual decay in *both* training and test images error as $K$ ranges from $N$ to $2N$ (the overcomplete regime). To understand this behavior, note that the covariance matrices of image patches *approximately* share an eigen-decomposition [36]. To see this, let $\mathbf{Q}(c)$ denote the orthogonal matrix of eigenvectors corresponding to the context-dependent covariance matrix $\mathbf{C}_{ss}(c)$. As shown in Fig. 4F (green histogram), there is a small, but non-negligible, difference between the eigenvectors $\mathbf{Q}(c)$ and the eigenvectors $\mathbf{Q}_*$ of the *average* covariance matrix $\mathbb{E}_{c \sim p(c)}[\mathbf{C}_{ss}(c)]$. When $K = N$, the column vectors of $\mathbf{W}_T$ learn to align with $\mathbf{Q}_*$ (as shown in Fig. 4F, blue histogram), and the circuit *approximately* adaptively whitens the context-dependent stimulus inputs via gain modulation. As $K$ ranges from 1 to $N$, $\mathbf{W}_T$ progressively learns the eigenvectors of $\mathbf{Q}_*$ (Appx. D). Since $\mathbf{W}_T$ achieves a full set of eigenvectors at $K = N$, this results in a large drop in error when measured using the operator norm. Finally, as mentioned, there is a non-negligible difference between the eigenvectors $\mathbf{Q}(c)$ and the eigenvectors $\mathbf{Q}_*$. Therefore, increasing the number of interneurons from $N$ to $2N$ allows the circuit to learn an overcomplete set of basis vectors $\mathbf{W}_T$ to account for the small deviations between $\mathbf{Q}(c)$ and $\mathbf{Q}_*$, resulting in improved whitening error (Appx. D).

## 6  Discussion

We've derived an adaptive statistical whitening circuit from a novel objective (Eq. 3) in which the (inverse) whitening matrix is factorized into components that are optimized at different timescales. This model draws inspiration from the extensive neuroscience literature on rapid gain modulation [25] and long-term synaptic plasticity [17], and concretely proposes complementary roles for these computations: synaptic plasticity facilitates learning features that are invariant *across* statistical contexts while gain modulation facilitates adaptation *within* a statistical context. Experimental support for this will come from detailed understanding of natural sensory statistics across statistical contexts and estimates of (changes in) synaptic connectivity from wiring diagrams (e.g., [37]) or neural activities (e.g., [38]).

Our circuit uses local learning rules for the gain and synaptic weight updates that could potentially be implemented in low-power neuromorphic hardware [39] and incorporated into existing mechanistic models of neural circuits with whitened or decorrelated responses [40–45]. However, there are aspects of our circuit that are not biologically realistic. For example, we do not sign-constrain the gains or synaptic weight matrices, so our circuit can violate Dale's law. In addition, the feedforward synaptic weights $\mathbf{W}^\top$ and feedback weights $-\mathbf{W}$ are constrained to be symmetric. In Appx. E, we describe modifications of our model that can make it more biologically realistic. Additionally, while we focus on the potential joint function of gain modulation and synaptic plasticity in adaptation, short-term synaptic plasticity, which operates on similar timescales as gain modulation, has also been reported [18]. Theoretical studies suggest that short-term synaptic plasticity is useful in multi-timescale learning tasks [46–48] and it may also contribute to multi-timescale adaptive whitening. Ultimately, support for different adaptation mechanisms will be adjudicated by experimental observations.

Our work may also be relevant beyond the biological setting. Decorrelation and whitening transformations are common preprocessing steps in statistical and machine learning methods [49–53], and are useful for preventing representational collapse in recent self-supervised learning methods [54–57]. Therefore, our online multi-timescale algorithm may be useful for developing adaptive self-supervised learning algorithms. In addition, our work is related to the general problem of online meta-learning [58; 59]; that is, learning methods that can rapidly adapt to new tasks. Our solution—which is closely related to mechanisms of test-time feature gain modulation developed for machine learning models for denoising [28], compression [60; 61], and classification [62]—suggests a general approach to meta-learning inspired by neuroscience: structural properties of the tasks (contexts) are encoded in synaptic weights and adaptation to the current task (context) is achieved by adjusting the gains of individual neurons.

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

## A  Notation

For $N, K \geq 2$, let $\mathbb{R}^N$ denote $N$-dimensional Euclidean space equipped with the usual Euclidean norm $\| \cdot \|$ and let $\mathbb{R}_+^N$ denote the non-negative orthant. Let $\mathbf{0} = [0, \ldots, 0]^\top$ and $\mathbf{1} = [1, \ldots, 1]^\top$ respectively denote the vectors of zeros and ones, whose dimensions should be clear from context.

Let $\mathbb{R}^{N \times K}$ denote the set of $N \times K$ real-valued matrices. Let $\| \cdot \|_F$ denote the Frobenius norm and $\| \cdot \|_{\text{op}}$ denote the operator norm. Let $O(N)$ denote the set of $N \times N$ orthogonal matrices. Let $\mathbb{S}^N$ (resp. $\mathbb{S}_{++}^N$) denote the set of $N \times N$ symmetric (resp. positive definite) matrices. Let $\mathbf{I}_N$ denote the $N \times N$ identity matrix.

Given vectors $\mathbf{u}, \mathbf{v} \in \mathbb{R}^N$, let $\mathbf{u} \circ \mathbf{v} = [u_1 v_1, \ldots, u_N v_N]^\top \in \mathbb{R}^N$ denote the Hadamard (elementwise) product of $\mathbf{u}$ and $\mathbf{v}$. Let $\text{diag}(\mathbf{u})$ denote the $N \times N$ diagonal matrix whose $(i, i)^{\text{th}}$ entry is $u_i$. Given a matrix $\mathbf{M} \in \mathbb{R}^{N \times N}$, we also let $\text{diag}(\mathbf{M})$ denote the $N$-dimensional vector whose $i^{\text{th}}$ entry is $M_{ii}$.

## B  Calculation details

### B.1  Minimum of the objective

Here we show that the minimum of $f_c$, defined in equation 2, is achieved at $\mathbf{M}_c = \mathbf{C}_{ss}^{1/2}(c)$. Differentiating $f_c(\mathbf{M})$ with respect to $\mathbf{M}$ yields

$$\nabla_{\mathbf{M}} f_c(\mathbf{M}) = -\mathbf{M}^{-1}\mathbf{C}_{ss}(c)\mathbf{M}^{-1} + \mathbf{I}_N. \tag{10}$$

Setting the gradient to zero and solving for $\mathbf{M}$ yields $\mathbf{M} = \mathbf{C}_{ss}^{1/2}(c)$. Substituting into $f_c$ yields $f_c(\mathbf{M}_c) = \text{Tr}(\mathbf{M}_c^{-1/2}\mathbf{C}_{ss}(c) + \mathbf{M}_c) = 2\,\text{Tr}(\mathbf{M}_c)$.

## B.2 Adding neural responses

Here we show that equation 4 holds. First, note that the trace term in equation 4 is strictly concave with respect to $\mathbf{r}$ (assuming $\mathbf{M}$ is positive definite) and setting the derivative equal to zero yields

$$2\mathbf{s} - 2\mathbf{M}\mathbf{r} = 0.$$

Therefore, the maximum in equation 4 is achieved at $\mathbf{r} = \mathbf{M}^{-1}\mathbf{s}$. Substituting into equation 4 with this form for $\mathbf{r}$, we get

$$
\begin{aligned}
\mathbb{E}_{\mathbf{s} \sim p(\mathbf{s}|c)} \left[ \max_{\mathbf{r} \in \mathbb{R}^N} \mathrm{Tr} \left( 2\mathbf{r}\mathbf{s}^\top - \mathbf{M}\mathbf{r}\mathbf{r}^\top + \mathbf{M} \right) \right] &= \mathbb{E}_{\mathbf{s} \sim p(\mathbf{s}|c)} \left[ \mathrm{Tr} \left( \mathbf{M}^{-1}\mathbf{s}\mathbf{s}^\top + \mathbf{M} \right) \right] \\
&= \mathrm{Tr} \left( \mathbf{M}^{-1}\mathbf{C}_{ss}(c) + \mathbf{M} \right) \\
&= f_c(\mathbf{M}),
\end{aligned}
$$

where the second equality uses the linearity of the expectation and trace operators as well as the formula $\mathbf{C}_{ss}(c) := \mathbb{E}_{\mathbf{s} \sim p(\mathbf{s}|c)}[\mathbf{s}\mathbf{s}^\top]$. This completes the proof that equation 4 holds. Next, using this expression for $f_c$, we have

$$
\begin{aligned}
f_c \left( \alpha \mathbf{I}_N + \mathbf{W}\mathrm{diag}(\mathbf{g})\mathbf{W}^\top \right) &= \mathbb{E}_{\mathbf{s} \sim p(\mathbf{s}|c)} \left[ \max_{\mathbf{r} \in \mathbb{R}^N} \mathrm{Tr} \left( 2\mathbf{r}\mathbf{s}^\top - \alpha \mathbf{r}\mathbf{r}^\top - \mathbf{W}\mathrm{diag}(\mathbf{g})\mathbf{W}^\top \mathbf{r}\mathbf{r}^\top \right) \right] \\
&\quad + \alpha N + \mathrm{Tr}(\mathbf{W}\mathrm{diag}(\mathbf{g})\mathbf{W}^\top) \\
&= \mathbb{E}_{\mathbf{s} \sim p(\mathbf{s}|c)} \left[ \max_{\mathbf{r} \in \mathbb{R}^N} \ell(\mathbf{W}, \mathbf{g}, \mathbf{r}, \mathbf{s}) \right] + \alpha N.
\end{aligned}
$$

Substituting into equation 3 and dropping the constant $\alpha N$ term results in equation 5.

## C   Offline multi-timescale adaptive whitening algorithm

We consider an algorithm where we directly optimize the objective in equation 3. In this case, we assume that the input to the algorithm is a sequence of covariance matrices $\mathbf{C}_{ss}(1), \mathbf{C}_{ss}(2), \ldots$. Within each context $c = 1, 2, \ldots$, we take $J \geq 1$ concurrent gradient descent steps with respect to $\mathbf{g}$ and $\mathbf{W}$:

$$
\begin{aligned}
\Delta \mathbf{g} &= -\eta_g \mathrm{diag} \left[ \mathbf{W}^\top \nabla_\mathbf{M} f_c(\alpha \mathbf{I}_N + \mathbf{W}\mathrm{diag}(\mathbf{g})\mathbf{W}^\top)\mathbf{W} \right], \\
\Delta \mathbf{W} &= -\eta_w \nabla_\mathbf{M} f_c(\alpha \mathbf{I}_N + \mathbf{W}\mathrm{diag}(\mathbf{g})\mathbf{W}^\top)\mathbf{W}\mathrm{diag}(\mathbf{g}),
\end{aligned}
$$

where we assume $\eta_g \gg \eta_w > 0$ as in Algorithm 1 and the gradient of $f_c(\mathbf{M})$ with respect to $\mathbf{M}$ is given in equation 10. These updates for $\mathbf{g}$ and $\mathbf{W}$ can also be obtained by averaging the corresponding updates in equations 7 and 8 over the conditional distribution $p(\mathbf{s}|c)$. This results in Algorithm 2.

---

**Algorithm 2:** Offline multi-timescale adaptive whitening

1: **Input:** Covariance matrices $\mathbf{C}_{ss}(1), \mathbf{C}_{ss}(2), \ldots$
2: **Initialize:** $\mathbf{W} \in \mathbb{R}^{N \times K}$; $\mathbf{g} \in \mathbb{R}^K$; $\alpha \geq 0$; $J \geq 1$; $\eta_g \gg \eta_w > 0$
3: **for** $c = 1, 2, \ldots$ **do**
4:     **for** $j = 1, \ldots, J$ **do**
5:         $\mathbf{M} \leftarrow \alpha \mathbf{I}_N + \mathbf{W}\mathrm{diag}(\mathbf{g})\mathbf{W}^\top$
6:         $\mathbf{g} \leftarrow \mathbf{g} - \eta_g \mathrm{diag}[\mathbf{W}^\top \nabla_\mathbf{M} f_c(\mathbf{M})\mathbf{W}]$
7:         $\mathbf{W} \leftarrow \mathbf{W} - \eta_w \nabla_\mathbf{M} f_c(\mathbf{M})\mathbf{W}\mathrm{diag}(\mathbf{g})$
8:     **end for**
9: **end for**

---

## D   Adaptive whitening of natural images

In this section, we elaborate on the converged structure of $\mathbf{W}_T$ using natural image patches. To better visualize the relationship between the learned columns of $\mathbf{W}$ and sinusoidal basis functions (e.g.

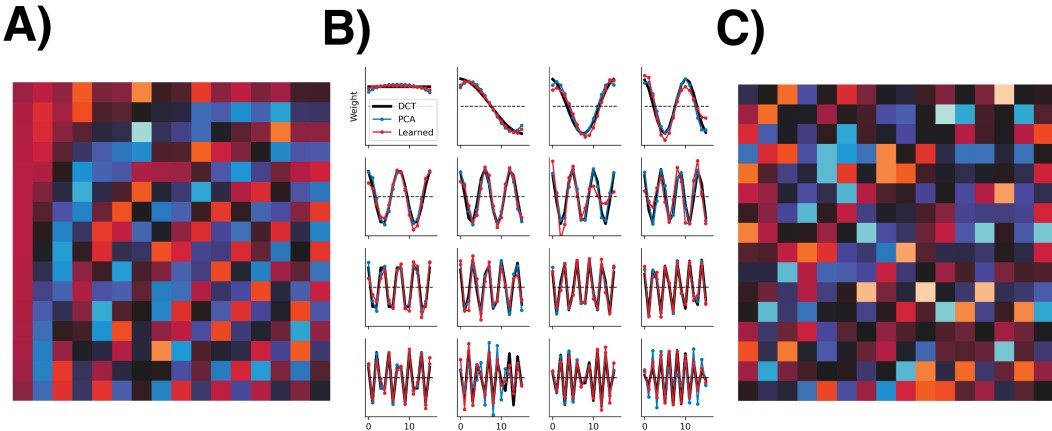

Figure 5: Control experiment accompanying Sec. 5.2. **A)** $\mathbf{W}_T$ learned from natural image patches. **B)** Basis vectors from **A** displayed as line plots, compared to the 1D DCT, and principal components of $\mathbb{E}_{c\sim p(c)}[\mathbf{C}_{ss}(c)]$. **C)** Control condition. $\mathbf{W}_T$ learned from spectrally-matched image patches with random eigenvectors.

DCT), we focus on 1-dimensional image patches (rows of pixels). The results are similar with 2D image patches.

It is well known that eigenvectors of natural images are well-approximated by sinusoidal basis functions [e.g. the DCT; 35; 36]. Using the same images from the main text [34], we generated 56 contexts by sampling $16 \times 1$ pixel patches from separate images, with 2E4 samples each. We train Algorithm 2 with $K = N = 16$, $\eta_w = 5\mathrm{E}{-2}$, and random $\mathbf{W}_0 \in O(16)$ on a training set of X of the images, presented uniformly at random $T = 1\mathrm{E}5$ times. Fig 5A,B shows that $\mathbf{W}_T$ approximates the principal components of the aggregated context-dependent covariance, $\mathbb{E}_{c\sim p(c)}[\mathbf{C}_{ss}(c)]$, which are closely aligned with the DCT. To show that this structure is inherent in the spatial statistics of natural images, we generated control contexts, $\mathbf{C}_{ss}(c)$, by forming covariance matrices with matching eigenspectra, but each with *random* and distinct eigenvectors. This destroys the structure induced by natural image statistics. Consequently, the learned vectors in $\mathbf{W}_T$ are no longer sinusoidal (Fig 5C). As a result, whitening error with $\mathbf{W}_T$ is much higher on the training set, with $0.3 \pm 0.02$ error (mean $\pm$ standard error over 10 random initializations; Eq. 9) on natural image contexts and $2.7 \pm 0.1$ on the control contexts. While for the natural images, a basis approximating the DCT was sufficient to adaptively whiten all contexts in the ensemble, this is not the case for the generated control contexts.

Finally, we find that as $K$ increases from $K = 1$ to $K = 16$, the basis vectors in $\mathbf{W}_T$ *progressively* learn higher frequency components of the DCT (Fig. 6). This is a sensible solution, due to the $\ell_2$ reconstruction error of our objective, and the $1/f$ spectral content of natural image statistics. With more flexibility, as $K$ increases past $N$ (i.e. the overcomplete regime), the network continues to improve its whitening error (Fig. 7A) by learning a basis, $\mathbf{W}_T$, that can account for within-context information that is insufficiently captured by the DCT (Fig. 7B). Taken together, our model successfully learns a basis $\mathbf{W}_T$ that exploits the spatial structure present in natural images.

# E    Modifications for increased biological realism

In this section, we modify Algorithm 1 to be more biologically realistic.

## E.1    Enforcing unit norm basis vectors

In our algorithm, there is no constraint on the magnitude of the column vectors of $\mathbf{W}$. We can enforce a unit norm (here measured using the Euclidean norm) constraint by adding Lagrange multipliers to the objective in equation 3:

$$\min_{\mathbf{W}\in\mathbb{R}^{N\times K}} \max_{\mathbf{m}\in\mathbb{R}^K} \mathbb{E}_{c\sim p(c)} \left[ \min_{\mathbf{g}\in\mathbb{R}^K} \mathbb{E}_{\mathbf{s}\sim p(\mathbf{s}|c)} \left[ g\left(\mathbf{W}, \mathbf{g}, \mathbf{r}, \mathbf{s}\right) \right] \right], \tag{11}$$

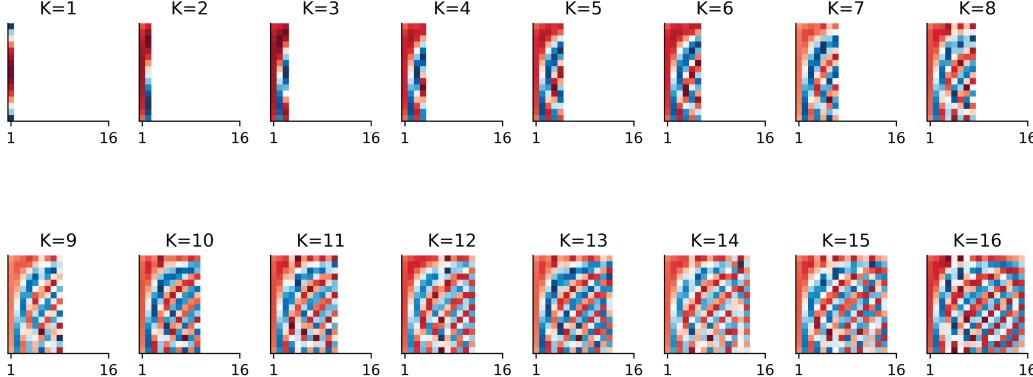

Figure 6: As $K$ increases, columns of $\mathbf{W}$ progressively learn higher frequency components of the DCT.

**A)**

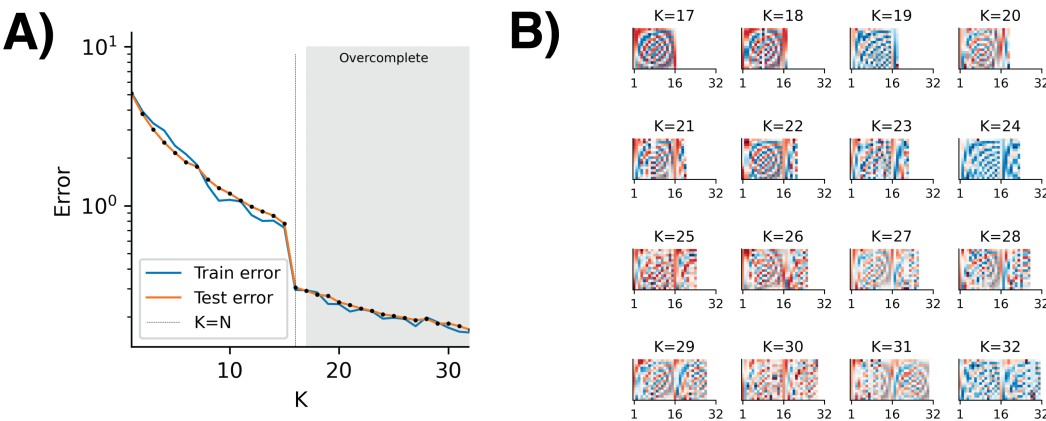

**B)**

Figure 7: **A)** Error on training and test set as a function of $K$. **B)** In the overcomplete regime, the network learns basis vectors $\mathbf{W}_T$ that further improve the error compared with the $K \leq N$ regime.

where

$$g(\mathbf{W}, \mathbf{g}, \mathbf{r}, \mathbf{s}) = \ell(\mathbf{W}, \mathbf{g}, \mathbf{r}, \mathbf{s}) + \sum_{i=1}^{K} m_i(\|\mathbf{w}_i\|^2 - 1).$$

Taking partial derivatives with respect to $\mathbf{w}_i$ and $\mathbf{m}_i$ results in the updates:

$$\Delta \mathbf{w}_i = \eta_w(n_i \mathbf{r} - (g_i + m_i)\mathbf{w}_i)$$
$$\Delta m_i = \|\mathbf{w}_i\|^2 - 1.$$

Furthermore, since the weights are constrained to have unit norm, we can replace $\|\mathbf{w}_i\|^2$ with 1 in the gain update:

$$\Delta g_i = \eta_g(z_i^2 - 1).$$

### E.2 Decoupling the feedforward and feedback weights

We replace the primary neuron-to-interneuron weight matrix $\mathbf{W}^\top$ (resp. interneuron-to-primary neuron weight matrix $-\mathbf{W}$) with $\mathbf{W}_{rn}$ (resp. $-\mathbf{W}_{nr}$). In this case, the update rules are

$$\mathbf{W}_{rn} \leftarrow \mathbf{W}_{rn} + \eta_w \left( \mathbf{n}_t \mathbf{r}_t^\top - \text{diag}(\mathbf{g} + \mathbf{m})\mathbf{W}_{rn} \right)$$
$$\mathbf{W}_{nr} \leftarrow \mathbf{W}_{nr} + \eta_w \left( \mathbf{r}_t \mathbf{n}_t^\top - \mathbf{W}_{nr}\text{diag}(\mathbf{g} + \mathbf{m}) \right).$$

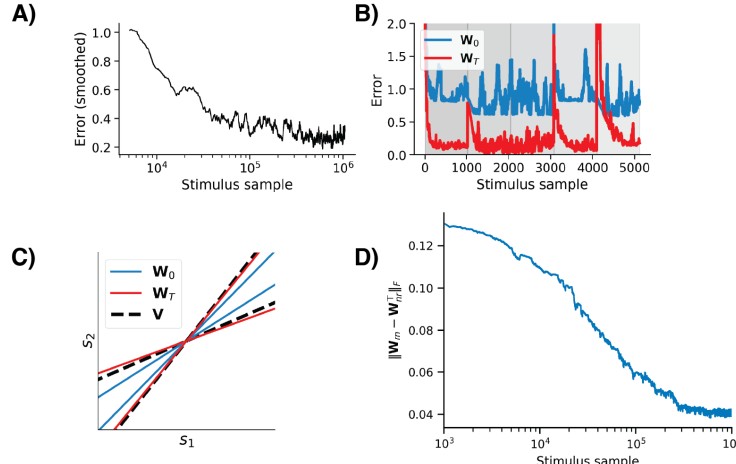

Figure 8: With stricter constraints on biological realism (Algorithm 3), the model succeeds to whiten synthetic data setup from Section 5.1 when the column vectors of $\mathbf{V}$ are chosen to be nonnegative. **A)** Error decreases as training progresses. **B)** Online algorithm performance before and after training. **C)** The learned basis (row vectors of $\mathbf{W}_{nr}$ after learning, red) is well-aligned with the generative basis (dashed black) compared to initialization (row vectors of $\mathbf{W}_{nr}$ at initialization, blue). **D)** Asymmetric feedforward weights, $\mathbf{W}_{rn}$, and feedback weights, $-\mathbf{W}_{nr}$, converge to being symmetric.

Let $\mathbf{W}_{rn,t}$ and $\mathbf{W}_{nr,t}$ denote the values of the weights $\mathbf{W}_{rn}$ and $\mathbf{W}_{nr}$, respectively, after $t = 0, 1, \ldots$ iterates. Then for all $t = 0, 1, \ldots$,

$$\mathbf{W}_{rn,t}^\top - \mathbf{W}_{nr,t} = \left( \mathbf{W}_{rn,0}^\top - \mathbf{W}_{nr,0} \right) \left( \mathbf{I}_N - \eta_w \mathrm{diag}(\mathbf{g} + \mathbf{m}) \right)^t.$$

Thus, if $g_i + m_i \in (0, 2\eta_w^{-1})$ for all $i$ (e.g., by enforcing non-negative $g_i, m_i$ and choosing $\eta_w > 0$ sufficiently small), then the difference decays exponentially in $t$ and the feedforward and feedback weights are asymptotically symmetric. This result is similar to that of Kolen and Pollack [63], where the authors trained a network with weight decay to show that weight symmetry naturally arises to induce bidirectional function synchronization during learning.

### E.3  Sign-constraining the synaptic weights and gains

The synaptic weight matrix $\mathbf{W}$ and gains vector $\mathbf{g}$ are not sign-constrained in Algorithm 1, which is not consistent with biological evidence. We can modify the algorithm to enforce the sign constraints by rectifying the weights and gains at each step. Here $[\cdot]_+$ denote the elementwise rectification operation. This results in the updates

$$\mathbf{g} \leftarrow \left[ \mathbf{g} + \eta_g (\mathbf{z} \circ \mathbf{z} - \mathbf{1}) \right]_+$$
$$\mathbf{W}_{rn} \leftarrow \left[ \mathbf{W}_{rn} + \eta_w \left( \mathbf{n}_t \mathbf{r}_t^\top - \mathrm{diag}(\mathbf{g} + \mathbf{m}) \mathbf{W}_{rn} \right) \right]_+$$
$$\mathbf{W}_{nr} \leftarrow \left[ \mathbf{W}_{nr} + \eta_w \left( \mathbf{r}_t \mathbf{n}_t^\top - \mathbf{W}_{nr} \mathrm{diag}(\mathbf{g} + \mathbf{m}) \right) \right]_+ .$$

### E.4  Online algorithm with improved biological realism

Combining these modifications yields our more biologically realistic multi-timescale online algorithm, Algorithm 3.

---

**Algorithm 3:** Biologically realistic multi-timescale adaptive whitening

---

1: **Input:** $\mathbf{s}_1, \mathbf{s}_2, \cdots \in \mathbb{R}^N$
2: **Initialize:** $\mathbf{W}_{nr} \in \mathbb{R}_+^{N \times K}$; $\mathbf{W}_{rn} \in \mathbb{R}_+^{K \times N}$; $\mathbf{m}, \mathbf{g} \in \mathbb{R}_+^K$; $\eta_r, \eta_m > 0$; $\eta_g \gg \eta_w > 0$
3: **for** $t = 1, 2, \ldots$ **do**
4:     $\mathbf{r}_t \leftarrow \mathbf{0}$
5:     **while** not converged **do**
6:        $\mathbf{z}_t \leftarrow \mathbf{W}_{rn}\mathbf{r}_t$ ;                                `// interneuron inputs`
7:        $\mathbf{n}_t \leftarrow \mathbf{g} \circ \mathbf{z}_t$ ;                     `// gain-modulated interneuron outputs`
8:        $\mathbf{r}_t \leftarrow \mathbf{r}_t + \eta_r \left( \mathbf{s}_t - \mathbf{W}_{nr}\mathbf{n}_t - \alpha\mathbf{r}_t \right)$ ;       `// recurrent neural dynamics`
9:     **end while**
10:    $\mathbf{m} \leftarrow \left[ \mathbf{m} + \eta_m (\text{diag}(\mathbf{W}_{rn}\mathbf{W}_{nr}) - \mathbf{1}) \right]_+$ ;     `// weight normalization update`
11:    $\mathbf{g} \leftarrow \left[ \mathbf{g} + \eta_g \left( \mathbf{z}_t \circ \mathbf{z}_t - \mathbf{1} \right) \right]_+$ ;                   `// gains update`
12:    $\mathbf{W}_{rn} \leftarrow \left[ \mathbf{W}_{rn} + \eta_w \left( \mathbf{n}_t\mathbf{r}_t^\top - \text{diag}(\mathbf{g} + \mathbf{m})\mathbf{W}_{rn} \right) \right]_+$ ;   `// synaptic weights update`
13:    $\mathbf{W}_{nr} \leftarrow \left[ \mathbf{W}_{nr} + \eta_w \left( \mathbf{r}_t\mathbf{n}_t^\top - \mathbf{W}_{nr}\text{diag}(\mathbf{g} + \mathbf{m}) \right) \right]_+$
14: **end for**

---

We test Algorithm 3 on a similar synthetic data setup to what was used in section 5.1, except that we sample the column vectors of $\mathbf{V}$ from the intersection of the unit circle with the nonnegative quadrant, Figure 8.

