# OpenReview forum: "Adaptive whitening with fast gain modulation and slow synaptic plasticity"
_NeurIPS.cc/2023/Conference — NeurIPS 2023 spotlight_

### Official Review · Reviewer_yS92 · 2023-06-26

**Soundness:** 3 good
**Presentation:** 2 fair
**Contribution:** 2 fair
**Rating:** 6
**Confidence:** 4

**Summary:**

This paper proposes a normative principle for the symmetric whitening problem, i.e., a batch optimization problem which is provable to attain symmetric whitening is used to derive an online adaptive algorithm. The proposed framework unifies the previous works, and the resulting algorithm maps into single layer network with interneurons and local learning rules. The mathematical findings of this normative approach is also supported with the evidences from neuroscience. The proposed approached is tested with synthetic data and natural images to demonstrate the performance. Potential improvements for more biologically realistic method is also discussed.

**Strengths:**

The paper provides a neural network solution to the whitening problem which satisfies two important constraints for biological plausibility: i) the network operates in an online manner, and ii) the weight and gain updates are local. The proposed online algorithm is illustrated to obtain whitening with both synthetic and natural datasets.

**Weaknesses:**

The writing is lacking in some parts, and some important details are missing in the paper. More numerical experiments should be presented to assess the introduced framework. For exact details on the weaknesses, please see my comments below and questions section.

   * Line 83, "... where $f_c(\mathbf{M}_c) = 2 \mathbf{M}_c$" should be changed to "... where $f_c(\mathbf{M}_c) = \text{Tr}(2 \mathbf{M}_c)$" since $f_c$ is a scalar function of matrices. Moreover, even though it is a simple proof, it would be valuable how this unique minimum is achieved.

   * The intermediate step (stated between line 150 and line 153) to obtain Equation (4) should be written 	explicitly in the paper (can be in the appendix).

   * The decoupling the feedforward and feedback weights (in Appendix D.2) is very similar to the idea presented in [i] since the assumption $g_i + m_i \in (0, 2 \eta_w^{-1})$ corresponds to weight decaying. The paper should cite this reference.

    [i]J.F. Kolen and J.B. Pollack. Backpropagation without weight transport. In Proceedings of 1994 IEEE International Conference on Neural Networks (ICNN’94), volume 3, pages 1375–1380 vol.3, 1994. doi: 10.1109/ICNN.1994.374486.

   * For the experiment with synthetic data, the experiment only considers the whitening matrix of the form $\mathbf{M}_c = \mathbf{I}_N + \mathbf{V} \mathbf{\Lambda}(c)\mathbf{V}^T$. An experiment with more general form of whitening matrix is required to assess the performance of the proposed algorithm.

   * Hyperparameters (i.e., $\eta_w, \eta_g, \eta_r$) are not provided in the paper (except that the authors provide $\eta_w$ for Algorithm 2 in Appendix C). It would be beneficial for the paper to include a table that provides the hyperparameters used in each experiment

   * The differences and novelties compared to previous works by Pehlevan and Chklovskii [11] and Duong et al. [18] are discussed quite well, but the comparison in the numerical experiments is missing.

   * A single Python script is shared for the proposed method, although it does not include the experiment code. The authors mentioned in the paper that the full code would be shared upon publication. However, the current code does not provide a comprehensive understanding of the conducted experiments.


**Questions:**

* The numerical experiments presented in Appendix C uses Algorithm 2, but the biological plausibility of it is questionable due to matrix inversion and batch learning. Did you experiment with this setup using Algorithm 1?

* The authors propose Algorithm 3 as more biologically plausible algorithm compared to the Algorithm 1, by complying to the Dale's law and decoupling the feedforward and feedback weights for asymmetry (weight transport problem). However, no numerical experiment has been demonstrated for this proposal. Did you test this algorithm?

* The value of $\alpha$ seems to be also an hyperparameter for the proposed framework. Did you analyze the effect of this choice?

**Limitations:**

I think that the limitations are adequately discussed in the paper.

---

> ### Author Rebuttal · Authors · 2023-08-09
>
> Thank you for your careful reading of our work and for your suggestions. We regret that you found the writing lacking in some parts. We have revised our paper in accordance with your suggestions, which we believe has improved the overall clarity of the paper.
>
> We are concerned that a central contribution of our work may not have been appreciated. Our primary motivation is to develop a biologically realistic circuit for adaptive, _context-dependent_ whitening with complementary computational roles for synaptic plasticity and gain modulation. We formalize existing mechanisms of gain modulation and synaptic plasticity into a joint computation operating across multiple timescales, yielding a new mechanistic theory and solution to context-dependent adaptation in neural circuits. We believe this is a novel and significant contribution, which we will emphasize in our revision.
>
> Weaknesses:
>
> 1. We corrected the equation and added a short proof that explains how the minimum is achieved.
>
> 2. We added a short section to the appendix with the intermediate steps written explicitly.
>
> 3. Thank you for pointing us to reference [i]. It is indeed relevant and we've added a citation.
>
> 4. A strength of our normative approach is that its performance can be predicted by analyzing the starting objective. In particular, given ${\bf W}$, we can analytically describe the set of covariance matrices that can be whitened by the circuit (see the display after eq 3). Our numerical simulations verify this analytical prediction. Given the analytical nature of our results, we believe that the current set of experiments adequately validates the performance of the circuit.
>
> 5. We amended the text to include a table of these hyperparameters.
>
> 6. The Pehlevan and Chklovskii [11] and Duong et al. [18] models, which respectively use either synaptic plasticity or gain modulation, correspond to our network in the regimes $\eta_g=0$ and $\eta_w=0$, respectively. A comparison to these regimes is shown in Fig. 4C, where we quantify the test error for our model (red histogram), for fixed gains [11] (green histogram) and for fixed synapses [18] (purple histogram). We amended the text to emphasize this comparison to previous work.
>
> 7. We have submitted full code (to the AC via an anonymous link, as instructed) for the synthetic data experiments.
>
> Questions:
>
> 1. Yes, using the Algorithm 1 for e.g. Appendix Fig C.1 still learns an approximately orthogonal sinusoidal basis for the data. The main difference is that the online algorithm requires far more iterations than the offline algorithm.
>
> 2. Yes, see Fig E.2 in the rebuttal PDF.
>
> 3. Yes, we've analyzed the $\alpha$ parameter, which corresponds to the leak term in the neural dynamics. In the context of the synthetic experiment when the generative basis, ${\bf V}$, is not orthogonal, a non-zero $\alpha$ parameter is necessary (effectively due to the matrix inversion); when ${\bf V}$ is orthogonal, $\alpha$ is less important and can be set to zero. See Fig E.3 in rebuttal PDF.

---

> > ### Comment · Reviewer_yS92 · 2023-08-10
> > **Response to the Rebuttal by Authors**
> >
> > I would like thank to the authors for their thoughtful responses. I genuinely appreciate the central contributions made by this work. My intention in offering my comments was to provide constructive insights on aspects that, in my opinion, could be beneficial for further refinement. In this context, I find that this rebuttal and global rebuttal address the concerns and queries I had raised.
> >
> > I think that the promised revisions and provided clarifications have already enhanced or will further enhance the current manuscript. I would like to also thank for the supplementary experiments presented in the global PDF, which are valuable additions to the work.
> >
> > As a result of these considerations, I am inclined to adjust my rating from 5 to 6.

---

### Official Review · Reviewer_NSc1 · 2023-07-06

**Soundness:** 3 good
**Presentation:** 4 excellent
**Contribution:** 2 fair
**Rating:** 7
**Confidence:** 4

**Summary:**

This paper gives an algorithm for learning weights of a neural network over long time scales, which allow interneurons to decorrelate the responses of excitatory neurons by modulating their gains over short time scales. On both synthetic and natural image datasets, this algorithm is shown to be effective and to generalize to new contexts.

**Strengths:**

* This paper makes progress on a fundamental question in neuroscience: How a population of neurons can coordinate their responses to stimuli to encode it. This work should also be of interest in machine learning, as it has obvious applications to transfer learning.
* Overall, the paper is clear and easy to understand.
* The learning algorithm is online and local.
* The technical evaluations are fairly thorough and support the paper's claims.

**Weaknesses:**

Some weaknesses are inherited from the Duong et al. model which this paper is building on, specifically related to its biological plausibility:
* All neurons are linear.
* Dale's law is not observed.
* Forward and backward weights are mirrored.
* In general, more interneurons than excitatory neurons are required (in the worst case, as many as $\Omega(n^2)$ interneurons for $n$ excitatory neurons), which does not match the ratios observed in the brain. Experimentally, the algorithm fails gracefully when fewer interneurons than required are available.
A more realistic version of the model which addresses the first two issues is provided in the appendix. However, given that the scope of this work is primarily modeling a phenomenon in neuroscience, I think the paper would be stronger if this biologically plausible version were featured and evaluated more prominently.

**Questions:**

* Line 83: Missing a trace on the RHS of the equation?
* I would appreciate a more in-depth evaluation, perhaps on a synthetic dataset, of what happens when there are fewer interneurons than required.
* Did you consider adding an activation function, at least to the interneurons? I could imagine that there is a similar learning algorithm which allows for fewer nonlinear interneurons.

**Limitations:**

Yes.

---

> ### Author Rebuttal · Authors · 2023-08-09
>
> Thank you for your careful review and for your useful comments. We appreciate your comment that our model has broader implications for ML and applications to transfer learning.
>
> Weaknesses: We readily acknowledge there are aspects of our model that are not biologically realistic. This is in part due to the fact that the model is derived from an objective that can whiten any input distribution, which is perhaps unrealistic for a biological system. While we did not have space to fully address this in the main paper, in Algorithm 3 of Appendix D, we considered modifications of our circuit to improve its biological realism.
>
> 1. Adding a nonlinearity is certainly worth pursuing in a future study. We will update an included reference on related work by Chapochnikov et al. [6] containing analyses on rectifying interneuron responses.
>
> 2. In Appendix D.2 we decouple the feedforward and feedback weight updates and in Appendix D.3 we consider a modified synaptic update rule that enforces Dale's law. In Figure E.2 of the attached PDF, we show that the modified algorithm indeed works on synthetic datasets.
>
> 3. See #2 above.
>
> 4. First, interneurons outnumbering excitatory neurons is in fact a relevant phenomenon observed in neuroscience, and exists in structures such as the olfactory bulb, where interneurons outnumber primary neurons by up to 100:1 [1]. Second, in Figure E.1 of the attached PDF, we show that even when $K\ll N$, the circuit responses which be much more decorrelated than the circuit inputs. This approximate decorrelation using fewer interneurons can perhaps explain why responses in V1 are only approximately decorrelated [3].
>
> Questions:
>
> 1. Thanks for pointing this out. We have corrected this equation.
>
> 2. In Fig 4E (natural images example) and Appendix C, we carefully analyze the effect of having fewer interneurons than required. In Fig 4D, we show that the "whitening error" gradually decreases with the number of interneurons $K$ until $K=N$, at which point there is a discontinuous decrease that is due to the fact that we are measuring the difference using the operator norm. We see that even for $K\approx N/3$, which corresponds to the ratio of interneurons to primary neurons in the cortex, the whitening error is much smaller than that for $K=1$. We have now also included the case with no interneurons ($K=0$) and therefore no whitening - see Figure E.1 of the attached pdf.
>
> 3. This is great suggestion for an extension to the model! Adding a nonlinear activation to the interneuron activations may indeed allow for a smaller number of interneurons $K$. Recent related work by Chapochnikov et al. [6] analyzes the role of rectifying interneurons in the whitening olfactory circuit in the context of similarity-matching-based adaptation.
> Here, we were focused on introducing the novel concept of multi-timescale whitening, and wish to keep the computation as analytically tractable as possible with this added layer of complexity. Any definitive claims on the whitening capability of our network with nonlinearities warrants a separate, follow-up study with extensive numerical/empirical investigation.
>
> [1] Shepherd, G. M. (Ed.). (2003). *The Synaptic Organization of the Brain*. Oxford University Press.
>
> [3] Benucci, A., Saleem, A. B., & Carandini, M. (2013). Adaptation maintains population homeostasis in primary visual cortex. *Nature Neuroscience*, 16(6), 724-729.
>
> [6] Chapochnikov, N. M., Pehlevan, C., & Chklovskii, D. B. (2023). Normative and mechanistic model of an adaptive circuit for efficient encoding and feature extraction. *Proceedings of the National Academy of Sciences*, 120(29), e2117484120.

---

> > ### Comment · Reviewer_NSc1 · 2023-08-13
> >
> > Thank you for your detailed response! I maintain that this is a good paper and stand by my original score.

---

### Official Review · Reviewer_3ayL · 2023-07-07

**Soundness:** 4 excellent
**Presentation:** 4 excellent
**Contribution:** 3 good
**Rating:** 7
**Confidence:** 5

**Summary:**

The authors produce a mechanistic model that combines synaptic plasticity and gain modulation to adaptively whiten responses. This model is constructed from an objective for learning a whitening transformation and then considering matrix factorization.  Simple factorization introduces interneurons and optimization via gradient descent can be mapped to synaptic plasticity which adapts the weights to a particular context.  Another decomposition that enforces a particular diagonalization leads whitening via gain modulation when the weights of the interneurons are fixed.  The contribution of this paper is to combine these two mechanisms into a single cost function and network instantiation in which the gain optimization produces a whitening for a given context, and the synaptic plasticity adapts to properties useful over multiple contexts.  The describe an implementation of this optimization in a recurrent neural network and consider different time scales, in which gain adaptation happens quickly within a context and synaptic plasticity happens over a longer timescale.  The authors test this framework on simulated data that matches the decomposition assumptions and show that the method is successful.  They further test it on natural images and show that the learned connectivity successfully adapts to new contexts.

**Strengths:**

This is a very nice paper that connects the computational problem of whitening with multiple neural mechanisms into a single normative framework.  While it still has issues with full biological plausibility (as noted by the authors) this is an interesting step in thinking about this computation and the roles of these neural mechanisms.  The presentation is very clear (Figure 2 is particularly nice).


**Weaknesses:**

There are no significant weaknesses in this paper.  To the authors’ credit, every question I had that I thought was in the category of “necessary for publication” they addressed.  I’ll leave everything else to “Questions”, below.

The biggest weakness I see regards the generality of the matrix decomposition.  The extent to which the authors test or address this appears on par with other approaches in the field, though.  I have some questions below on this topic.


**Questions:**

Can you make any statements about how general the matrix decomposition in Section 3.3 is?  It seems very specific.  You demonstrate numerically that this decomposition works for your experiment.  When do you expect it to fail?  Can you characterize or quantify the differences in context necessary before this decomposition no longer works?

Related to this, it looks like you’re using images from the van Hateren database.  Can you try the held out image experiments with images from very different contexts and origins, since those images all have clear similarities in large scale properties?  This might help find the limits of this algorithm/decomposition.

I was going to ask about breaking the symmetry on the weights, which you answered in an appendix.  They will asymptotically align, but can you say something about this timescale?  Is there a restriction on when the alignment has to happen with respect to the gain modulation and synaptic plasticity that is meaningful or prohibitive?

You provide the error as a function of number of interneurons for your natural image experiment.  Can you compute some normalization so that there is an idea of what errors are meaningful (maybe the Frobenius norm of C)?  Your interneurons are not necessarily the “interneurons” people talk about in real circuits, but it makes one imagine nonetheless that you will have far fewer interneurons than primary neurons.  Is this a functional problem given the amount of error you see?  Maybe it’s “good enough” even with few interneurons?

**Limitations:**

The authors do not discuss societal impact.

---

> ### Author Rebuttal · Authors · 2023-08-09
>
> We appreciate your positive review and thoughtful questions about the work!
>
> Questions:
>
> 1. We can be precise in describing the set of covariance matrices that can be whitened with this decomposition. In particular, when the inverse whitening matrices lie in a $K$-dimensional linear subspace, then this representation can match the whitening matrices with $K$ interneurons. Since the covariance matrices of natural images approximately share an eigen-basis, this representation can approximately match the corresponding whitening matrices with $K=N$. When the inverse whitening matrices do not lie in a low-dimensional linear subspace, we expect that the representation will require $K$ to be much larger, which can perhaps explain why some circuits have many more interneurons than primary neurons (e.g., olfactory bulb).
>
> 2. This is an interesting question which can be answered by appealing to well-known results from the image/video coding literature: a sinusoidal basis (e.g. the DCT) approximates the Principal components (Karhunen Loeve Transform) for natural images [4,5]. Our algorithm learns an approximately orthogonal sinusoidal basis as interneuron weights ${\bf W}$ without supervision (Fig 4D, 4F, C.1B, C.2); therefore, this trained model is well-suited to any held-out test images (e.g., outside the van Hateren database) whose statistics do not deviate too far from those of natural images.
>
> 3. Yes, as we show in Appendix D.2 that the convergence to symmetric weights is exponential. If the weights are order 1, then the exponential convergence rate is on the order $-\log(1-\eta_w)\approx\eta_w$ when $\eta_w$ is small. That is, the convergence rate is determined by the learning rate for the synaptic weight updates.
>
> 4. Thanks for pointing out this issue of interpretability of the error plots. To provide a better sense of error improvement, we've added a horizontal reference line to show what the error would be with no interneurons (i.e. without whitening).
>
> 5.  For olfactory bulb, it seems that this proportion of interneurons is in fact a relevant scenario: interneurons can outnumber excitatory neurons by up to 100:1 [1].
> We do show in Fig 4E and Appendix C that our network transitions gradually to the regime where $K\ll N$. In this regime when $K<N$, the error is indeed significantly improved (relative to no whitening --- horizontal line in PDF Fig E.1).
>
> [1] Shepherd, G. M. (Ed.). (2003). *The Synaptic Organization of the Brain*. Oxford University Press.
>
> [4] Ahmed, N., Natarajan, T., & Rao, K. R. (1974). Discrete cosine transform. *IEEE Transactions on Computers*, 100(1), 90-93.
>
> [5] Bull, D., & Zhang, F. (2021). *Intelligent image and video compression: communicating pictures.* Academic Press.

---

> > ### Comment · Reviewer_3ayL · 2023-08-20
> >
> > Thanks for the response.  I really enjoyed the paper.  I will maintain my score.

---

### Official Review · Reviewer_yTCH · 2023-07-10

**Soundness:** 3 good
**Presentation:** 3 good
**Contribution:** 3 good
**Rating:** 7
**Confidence:** 3

**Summary:**

This paper proposes a neural circuit model that combines fast gain modulation and slow synaptic plasticity to adaptively white sensory inputs. It appears that this paper is a combination of the studies of ref. 11 and 18.

**Strengths:**

### Originality
The strength of this paper is that it combines the fast gain modulation and slow synaptic plasticity in a whitening neural circuit, and addresses the shortcomings of earlier models with either gain modulation or synaptic plasticity. This is the most conceptual and technical advance of this study.

### Clarity
Overall, the paper is well written. But see my comments and questions on some parts I am unclear about.

**Weaknesses:**

I don't have major concerns about this study from the math point of view. Nonetheless, I have a significant concern from the neuroscience point of view.

The derived updating rule of gain (Eq. after line 174) depends on the current weight. The neural "gain" is typically adjusted by inhibitory interneurons, and thus I am concerned about how interneurons "sense" the synaptic weights. If there are neurobiological studies supporting this updating rule, please cite the reference in the paper, otherwise, it is better to discuss it at the end.

**Questions:**

- Line 91: ref. 11 requires $K \geq N$, i.e., the number of I neurons is larger than the E neurons. Does this requirement also need in this paper? Is there a way to make the number of I neurons less than the E neurons? This is also the case in the cortex.
- It seems that the $f_c(M)$ function was used before but the writing still looks abrupt to me about why to propose an objective function with such a form.
- The paragraph starting at line 90. It is better to first explain the decomposition $M_c = W_c W_c^T$ regards to the weights between E and I neurons. Otherwise, people can easily regard $W_c$ as the feedforward weights.

**Limitations:**

I realize the framework assumes the synaptic weights between excitatory (E) and inhibitory (I) neurons are symmetric with each other by just differing a sign. Is there a way to break this symmetry? Will the symmetry breaking of E-I weights facilitate optimization?

---

> ### Author Rebuttal · Authors · 2023-08-09
>
> Thank you for your careful review and for your thoughtful questions.
>
> Weaknesses:
>
> Thank you for voicing this concern. Although it's not so apparent in the main text,  we've resolved this issue in Appdx D.1 . Specifically, we scale the column vectors of ${\bf W}$ to have constant norm 1. This effectively replaces the $\text{diag}({\bf W}^\top {\bf W})$ term with a constant vector of ones, so gain updates do not need to "sense" the synaptic weights. This is reminiscent of the synaptic scaling/synaptic redistribution mechanisms discussed in Abbott \& Nelson [2].
> Additionally, when the weights are near-optimal, then they are approximately constant across contexts and the target term is effectively constant.
>
> Questions:
>
> 1. Yes, for cases such as in the cortex when $K<N$, it's still possible to produce whitened responses provided the set of (inverse) whitening matrices lie within a $K$-dimensional subspace. Further, some experiments in cortex have shown that primary neurons reduce redundancy after adaptation, but are not perfectly whitened [3]. The partially whitened solution found when $K<N$ may provide an explanation for this effect.
>
> 2. Thanks for pointing this out, we will amend the text as you suggest to improve clarity.
>
> 3. Thanks, we will make this change in conjunction with the previous one.
>
> Limitations:
>
> We discuss weight decoupling in Appendix D.2. But it is worth mentioning that reciprocal connections between excitatory and inhibitory neurons have been observed in structures such as the olfactory bulb [1], where dendrodendritic synapses give rise to symmetric connectivity matrices (although, not necessarily symmetric weight matrices). An exploration of circuit function with asymmetric weights (e.g. for non-symmetric whitening transforms) is an interesting direction to pursue. We have some preliminary findings suggest that non-symmetric weights can lead to non-symmetric whitening transformations.
>
> [1] Shepherd, G. M. (Ed.). (2003). *The Synaptic Organization of the Brain*. Oxford University Press.
>
> [2] Abbott, L. F., & Nelson, S. B. (2000). Synaptic plasticity: taming the beast. *Nature Neuroscience*, 3(11), 1178-1183.
>
> [3] Benucci, A., Saleem, A. B., & Carandini, M. (2013). Adaptation maintains population homeostasis in primary visual cortex. *Nature Neuroscience*, 16(6), 724-729.

---

> > ### Comment · Reviewer_yTCH · 2023-08-18
> >
> > Thanks authors' reply which addresses my concerns. Thus I increase my score from 6 to 7.

---

### Author Rebuttal · Authors · 2023-08-09

Thank you for your careful reading of our work and for your helpful comments. We have revised our paper in accordance with your suggestions and provide individual responses below. Here we list general changes and additions to the manuscript.

1. **Adaptation with fewer interneurons than primary neurons:** As the reviewers correctly pointed out, many circuits in the brain have far fewer interneurons than primary neurons (e.g., in the cortex the ratio is approximately 1:3). In Figure E.1 of the attached PDF (which is an updated version of Figure 4E from our original submission to include the case $K=0$), we show the performance of our algorithm on the images dataset when $K<N$. We see that even when there are relatively few interneurons (e.g., $K=4$), the whitening error is much less than when $K=0$. Perhaps neural circuits balance the benefits of reducing response correlations with the increased metabolic costs of having more interneurons. Finally, we mention that there are neural circuits in the brain where interneurons greatly outnumber primary neurons; e.g., in the olfactory bulb where the ratio of granule cells to mitral cells is upwards of 100:1 [1].
2. **Biological realism:** There are aspects of our model that are not biologically realistic. In our original submission, we briefly listed these aspects in the Discussion and in more depth in Appendix D, where we presented a more biologically realistic online algorithm (Algorithm 3).  We have now tested Algorithm 3 on a synthetic dataset and found that it successfully learned optimal filters and was able to adaptively whiten the data using gain modulation, see Figure E.2 of the attached PDF.
3. **Effect of hyperparameter $\alpha$:** We tested the effect of varying the $\alpha$ parameter on the synthetic dataset, Figure E.3 of the attached PDF. We find that when the column vectors of ${\bf V}$ are orthogonal, varying $\alpha$ does not affect the performance of the algorithm; however, when ${\bf V}$ are not orthogonal, there is a *slight* degradation of performance when $\alpha$ is not exactly 1. This is due to the fact that if ${\bf V}$ is orthogonal (and full rank), then $\alpha{\bf I}+{\bf V}\Lambda {\bf V}^\top$ can be decomposed as ${\bf V}(\alpha{\bf I}+\Lambda){\bf V}^\top$, so the basis vectors for the (inverse) whitening matrix do not depend on $\alpha$.

[1] Shepherd, G. M. (Ed.). (2003). *The Synaptic Organization of the Brain*. Oxford University Press.

---

### Decision · Program_Chairs · 2023-09-21

**Decision:**

Accept (spotlight)

**Comment:**

This paper examines the question of how neural circuits engage in whitening, i.e. normalizing the variance of responses and reducing correlations between responses. In this work, the authors build a model of neural whitening that uses fast gain adaptation and slow synaptic mechanisms. They test the algorithm on synthetic and natural datasets, and demonstrate that the synapses learn optimal solutions over long timescales that enable the circuit to adaptively whiten responses on short timescales using gain modulation.

The reviewers all agreed that this paper make an interesting, novel contribution and is overall well-written. The reviewers had some concerns related to the biological links and the tests run to compare the model, but the concerns were largely addressed by the rebuttals. Thus, this is a clear accept.